# Sleep Habits and Electronic Media Usage in Japanese Children: A Prospective Comparative Analysis of Preschoolers

**DOI:** 10.3390/ijerph17145189

**Published:** 2020-07-17

**Authors:** Fumie Horiuchi, Yasunori Oka, Kentaro Kawabe, Shu-ichi Ueno

**Affiliations:** 1Center for Child Health, Behavior and Development, Ehime University Hospital, Shitsukawa, 791-0295 Toon-City, Ehime prefecture, Japan; matsufu@m.ehime-u.ac.jp; 2Department of Neuropsychiatry, Ehime University Graduate School of Medicine, Shitsukawa, 791-0295 Toon-City, Japan; ueno@m.ehime-u.ac.jp; 3Center for Sleep Medicine, Ehime University Hospital, Shitsukawa, 791-0295 Toon-City, Japan; oka@sleepresearch.jp

**Keywords:** electronic media, sleep habits, portable games, home video games, kindergarten preschool children

## Abstract

Children are increasingly exposed to electronic media, which can potentially influence their sleep habits. However, few studies have investigated the effects of children’s life patterns on sleep habits and electronic media usage. This study investigated the differences in sleep habits and electronic media usage between 18- and 42-month-old children attending nursery schools, kindergartens, or staying at home, and respectively enrolled 183 (boys, *n* = 93; girls, *n* = 90) and 215 (boys, *n* = 104; girls, *n* = 111) 18- and 42-month-old children who underwent health check-ups. We found that 18-month-old children attending nursery school had significantly earlier wake times on weekdays and shorter sleep durations on weekends than children who stayed at home despite no differences in electronic media usage. There were no differences in sleep duration among 42-month-old children attending nursery schools, kindergartens, or staying at home; however, kindergarteners demonstrated a higher use of portable and home video games. Different life patterns affect electronic media usage in preschool children, especially those attending kindergarten. Particular attention should be paid to the higher usage of electronic media devices by kindergarteners, although they had the same sleep duration, as did other preschool children.

## 1. Introduction

Sleep is vital for physical and psychological well-being, especially in infants and children [1]. Sleep-wake regulation and sleep states evolve rapidly during the first year of life and continue to mature throughout childhood. The circadian rhythm begins to emerge at approximately 10–12 weeks of age, after which infant sleep becomes predominantly nocturnal [2]. Children aged between 1 and 4 years continue to take daytime naps to achieve their sleep requirements; however, on reaching 5 years of age, daytime napping ceases and, concurrently, overnight sleep duration gradually declines throughout childhood [3]. However, sleep-wake patterns vary widely and are driven by a complex interplay between biological processes and environmental, behavioral, and social factors.

Over the last decade, there has been a sharp increase in the availability and use of electronic media devices, such as smartphones, tablet devices, portable games, home video games, and computers, which have all had a strong influence on children’s lives. It has been reported that American children and adolescents aged between 8 and 18 years spend an average of 7 h a day on entertainment media, including television, computers, and handheld and other electronic devices [4]. This has led to less time being spent on homework, sleep, and play [5]. Moreover, the increased exposure to electronic media devices can affect the lifestyles of parents and caregivers. Additionally, evening exposure to bright light from a television or computer screen may suppress melatonin and consequently disrupt the circadian rhythm [6]. Vijaisakkhana et al. reported that, in infants at 12 months of age, screen exposure in the evening affected nighttime sleep duration [7].

There is a consensus among health authorities that excessive screen exposure time has an adverse effect on normal childhood development. The American Academy of Pediatrics guidelines recommend that children younger than 2 years should not spend time on electronic media, and media usage of children aged 2 or older should be restricted to <2 h per day [8,9]. Conversely, some parents consider media content to be educational, with one study reporting that 290 of 1000 parents interviewed allowed their children younger than 2 years to watch television because it was “good for their brain” [10]. The remaining parents admitted to not limiting their children’s screen time to avoid conflict or social isolation, or to provide entertainment or distraction [11]. Exposure to screens tends to start from very early infancy despite certain negative side effects of electronic media use [12,13,14].

Furthermore, the degree of screen exposure is related to preschool children’s daily life patterns, for example, attending nursery schools, kindergartens, or staying at home. The life patterns of Japanese young children vary according to their parents’ lifestyle and the family structure. The meaning of the terms; “kindergarten” and “nursery school” varies across countries. In Japan, kindergarten is a legally positioned school and is considered more of an educational preparation as a preschool. Kindergartens enroll children between the ages of 3 and 5 years and childcare hours are about 4 h as education standard time. On the other hand, Japanese nursery school is positioned as a child welfare facility under the jurisdiction of the Ministry of Health, Labor, and Welfare, which is essentially for children aged between 0 and 5 years whose parents/caregivers work during the day. Standard childcare hours there are about 11 h on definition. Children who begin in nursery school generally remain in nursery school until they are 5 years old. In terms of kindergarten’s cost for childcare depends on each kindergarten policy. According to the survey of the Ministry of Education, Culture, Sports, Science, and Technology in 2016, the cost of public kindergarten was 230,000 JPY (equivalent to 2150 US dollars)/year, and that of private kindergarten was 480,000 JPY (equivalent to 4500 US dollars)/year on average [15]. The data from Ministry of Health, Labor and Welfare in 2012 showed general childcare fee of nursery school was between 240,000 JPY (equivalent to 2240 US dollars)/year and 360,000 JPY (equivalent to 3360 US dollars) yen/year [16]. There are not big differences of activities; the mixture of adult-led and child-led play. In terms of nap, there is no time to have a nap in kindergarten. Conversely, there are naptime habitually in nursery school although it is not an obligation. Some parents prefer their children to stay at home rather than attend nursery schools or kindergartens. Thus, children aged between 0 and 3 years either attend nursery school or remain at home, and children in the 3–5 age range attend nursery school or kindergarten, although some stay at home.

Despite the variety of life patterns among children, no study has investigated the association between life patterns, sleep habits, and electronic media usage in children. We hypothesized that the life patterns of children aged 0–3 and 3–5 years have different effects on children’s sleep habits and electronic media usage. This study aimed to elucidate and investigate the characteristics of sleep habits and electronic media usage in children aged 18 and 42 months with a special focus on the differences in their lifestyles, including attendance at nursery school, kindergarten, or staying at home.

## 2. Materials and Methods

### 2.1. Participants

This study was conducted in Toon city, Ehime Prefecture, Japan, which had a population of 34,600 and 234 births in 2015. We enrolled all the children who attended a health check-up at age 18 or 42 months between September 2016 and October 2017. Therefore, the participants were children aged 18 months born between February 2014 and September 2016 (18M group) and children aged 42 months born between November 2012 and May 2014 (42M group). Their parents/caregivers were provided a written explanation about the study and its purpose and were asked to complete questionnaires concerning their child’s sleep habits, sleep-related problems, electronic media usage, and electronic media use time before the check-up. The response rates were 72.1% and 76.8% for the 18- and 42-month-old groups, respectively. The study was approved by the Institutional Review Board (IRB no. 1607009) of our institute, and consent to participate was assumed by the submission of the completed questionnaires, which was also approved by the Institutional Review Board.

### 2.2. Measures

The questionnaire included basic information, such as the child’s age, sex, height, weight, medical history, number of siblings and order of birth, and whether the child attended nursery school or kindergarten. Children’s sleep habits, such as information on sleep duration at nighttime, bedtime, wake time and the duration of daytime naps on weekdays and weekends and days per week with naps, were assessed using the Child and Adolescent Sleep Checklist, which has been developed and validated for clinical and research purposes [17,18,19].

The percentages of children who utilized a television, a laptop/computer, a tablet device, a music player, a smartphone, a mobile phone, a portable game, and/or a home video game, even if only very occasionally during the past month, were assessed. In addition, the median time spent on each electronic media device per day was measured.

### 2.3. Sample Size Calculation

The sample size was calculated based on a multiple logistic regression model using G* power (Faul, Erdfelder, Buchner, and Lang) 3.1.9.2 software (Heinrich-Heine-Universität, Düsseldorf, Germany) [20]. A small size effect of 0.4, a significance level of alpha = 0.05, and a statistical power of 1 − β = 0.95 were considered. The necessary sample size was fulfilled in this study.

### 2.4. Statistical Analysis

The 18M group was subdivided into those attending and not attending nursery school (“nursery school” and “home” groups, respectively). The 42M group was subdivided into those attending kindergarten, nursery school, or staying at home (“kindergarten,” “nursery school,” and “home” groups, respectively). Number of children by gender was compared using chi-square test. Data are expressed as median and percentiles (25, 75). In terms of the 18M groups, age, sleep habits were compared between the two groups using a Mann-Whitney *U*-test, and percentages of children who utilized electronic media even if very occasionally were compared using Fisher’s exact test. In the subgroup comparisons of the 42M group, Kruskal-Wallis test and Dunn’s test were undertaken to compare age, sex, and sleep habits. Percentages of children who utilized electronic media were assessed by the Fisher’s exact test. IBM SPSS Statistics version 22.0 (IBM Corporation, New York, United States) was used for statistical analysis, and the level of significance was set at *p* < 0.05.

## 3. Results

### 3.1. Demographic and Clinical Characteristics

Table 1 shows a comparison between the 18M and 42M groups with regard to participant characteristics and sleep habits. The 18M group comprised 183 children; 93 (50.8%) were boys, and 84 (45.9%) were firstborn children. The 42M group comprised 215 children; 104 (48.4%) were boys and 91 (42.3%) were firstborn children. The percentages of children who utilized electronic media even if very occasionally during the past month in the 18M and 42M groups are shown in Table 2.

### 3.2. A Comparison of Sleep Habits and Media Usage Between 18-Month-Old Children in the Nursery School and Home Groups

A subgroup comparison of the 18M groups—nursery school group (*n* = 79) and home group (*n* = 104)—is presented in Table 3.

There were no significant differences in sleep duration on weekdays or bedtime on weekdays and weekends between the two groups; however, the wake time on weekdays was significantly earlier in the nursery school group (*p* < 0.001), and the difference in wake time between weekdays and weekends was significantly greater in this group (*p* < 0.001). The nursery school group took longer naps than the home group (*p* < 0.05); however, there was no significant difference in the number of days in which the naps were taken. Means and percentiles (25, 75) of television use time were 60 min (15, 180 min) in Nursery school group, and 60 min (0, 180 min). Means of other electronic media use time were zero. There were no differences between the groups in the percentages of electronic media device usage (Table 4).

### 3.3. Comparison of Sleep Habits and Media Usage in 42-Month-Old Children Among Kindergarten, Nuersery School, and Home Subgroups

Table 5 and Table 6 shows a comparison of sleep habits and electronic media usage among the three subgroups of the 42M group: kindergarten group (*n* = 66), nursery school group (*n* = 114), and the home group (*n* = 35). 

Sleep duration on weekdays showed differences among three groups. Nursery school group had the longest sleep duration on weekdays, followed by Kindergarten and Home group. Bedtime on weekdays was significantly earlier for the kindergarten group than for the other groups, and bedtime at weekends was significantly earlier for the kindergarten group than for the nursery school group. On weekdays, wake time was significantly later in the home group than in the other groups; on weekends, wake time was significantly later in the home group than in the kindergarten group. The number of days per week on which naps were taken was significantly greater in the nursery school group, followed by the home group and the kindergarten group. Furthermore, the nap duration was significantly longer in the nursery school group than in the other groups. There were significant differences in electronic media usage, including the use of portable games (*p* = 0.003) and home video games (*p* = 0.033; Table 6). Medians and percentiles (25, 75) of television use time were 120 min (60, 180 min) in kindergarten group, 90 min (30, 172 min) in nursery school group, and 120 min (30, 225 min). Medians of other electronic media use time were zero. In the home group, 3.1% used portable games and 3.1% used home video games compared with 18.8% in the kindergarten group. In the nursery school group, 4.6% used portable games and 8.3% used home video games.

## 4. Discussion

This study compared sleep habits and electronic media usage in community–dwelling children aged 18 and 42 months, grouped according to their life patterns. Empirical research on the effects of exposure to electronic media devices and the influence of lifestyle patterns on electronic media usage in infants and toddlers is scarce. To our knowledge, this is the first study to compare sleep habits and media usage with regard to whether children attended kindergarten, nursery school, or stayed at home.

This study, conducted during health checkups, identified that the sleep habits of both 18- and 42-month-old children differed according to specific life patterns. Among the 18-month-old children, those who stayed at home had a longer sleep duration on weekends and they woke up later than those who attended nursery school. At 42 months, there was no statistically significant difference in weekday or weekend bedtime between nursery and home children; there was no statistically significant difference in weekday or weekend wake time between nursery and kindergarten children.

With regard to naps, there was a statistically significant difference in nap duration, not nap frequency between 18–month–old children attending nursery school and those staying at home. In contrast, significantly different patterns were observed in the 42-month-old children. In the 42M groups, children attending nursery school were found to nap on a greater number of days per week (6 days/week) and for a longer duration (90 min) than those in the other two groups. Fukuda et al. [21] reported the sleeping patterns of Japanese kindergarten and nursery school children in the age range of 3–6 years and found that nursery school children went to bed later at night and their night-time sleep duration was shorter than that of children who attended kindergarten. Moreover, the afternoon nap appeared to cause delayed sleep onset but was not a result of sleep deficiency, as shown through a comparison of the sleep duration on the previous night and the sleep-onset time between the days with and without an afternoon nap [21]. In this study, nap conditions significantly differed among the children in the 42M group attending kindergarten, attending nursery school, and staying at home. Especially nursery school children napped more frequently compared with children attending kindergarten or staying at home. It may be worth considering whether children attending nursery schools should have such long nap times.

Our research showed that television, smartphones, and tablet devices were the most popular electronic media devices used by both 18- and 42-month-old children, of whom 77% in the 18M group and 85% in the 42M group watched television regardless of their life pattern. The devices used most frequently by the 18M and the 42M group were smartphones (22% and 46%, respectively) and tablets (10% and 19%, respectively). The use of mobile devices, including smartphones and tablets, by young children has increased dramatically since the Kaiser Foundation began research into the use of technology for parents of 0- to 8-year-olds [22]. In 2011, 52% of children in the age range of 0–8 years were able to access mobile devices; however, by 2013, this had increased to 75% [23]. Recent trends indicate that smart devices are becoming increasingly popular among children. A touch–based multimodal interface smart device provides an easy-to-use platform for young children, especially when compared with using an electronic mouse, which requires fine motor and keyboard techniques [24].

There were no differences in electronic media usage between 18-month-old children attending nursey schools and those staying at home. However, a greater number of the 42M group attending kindergarten used portable games and home video games compared with those attending nursery schools or those staying at home. Children attending kindergarten arrive at kindergarten later and leave for home earlier than nursery school children. Consequently, kindergarten children would have more free time at home than those attending nursery schools. Moreover, it is possible that the 42M group attending kindergarten have more available time to be influenced by and/or taught how to work with electronic media devices by their elder siblings or by neighborhood children. However, it is unclear why more of the children remaining at home did not use electronic games. Our results indicated that more attention should be paid to children attending kindergartens, especially with regard to the acquisition of appropriate screen–time habits, because habits developed in childhood could have a significant effect throughout their lifetime.

Many studies have shown that excessive screen time for young children is associated with language delay, attention problems, obesity, aggressive behavior, and sleep problems [13,25,26,27]. Moreover, screen-time habits formed in early childhood have been shown to predict negative psychological and health outcomes later in life [28,29,30]. A recent longitudinal study demonstrated that parental monitoring of children’s media influenced their sleep, school performance, and prosocial and aggressive behaviors, and that limiting the amount of electronic media use and its content was a powerful protective factor for children between the third and fifth grades [31]. Our results indicating the use of screen media games in early childhood may indicate a need for early intervention. At present, it is up to parents to teach their children about media contents; however, parents themselves lack sufficient knowledge about the media and their influence on children. Parents must be taught neither to prohibit their children from using the media, nor should be controlled, but rather they should be accompanied [32]. Many opportunities should be prepared for not only children but also their parents to participate in media education programs from the early stage of children.

A strength of this study was that attention was paid to the children’s lifestyles rather than those of their caregivers. Moreover, sleep habits and electronic media usage were compared with regard to whether children attended kindergarten, nursery school, or stayed at home. This study had some limitations that should be considered for an appropriate interpretation of the results. First, the children’s sleep habits were evaluated by using questionnaires that were answered by the parents or caregivers, rather than through objective sleep measurements, such as actigraphy. Second, data on the parents’ socioeconomic status and behaviors were unavailable; therefore, this information could not be evaluated in relation to the children’s sleep patterns. Third, because of the relatively small sample size, we were unable to conduct comprehensive analyses on associations between children’s sleep habits, electronic media usage, and life patterns.

## 5. Conclusions

In summary, this study revealed that sleep habits and electronic media usage differed according to the age and life patterns of preschool children. A greater number of children attending kindergarten used portable games and home video games than the number of children attending nursery school or those who stayed at home. Therefore, it is essential to take into consideration children’s individual lifestyle patterns when advising families on appropriate electronic media use for their children. Further studies are needed to determine the most beneficial aspects of electronic media use in preschool children with regard to their lifestyle patterns.

## Figures and Tables

**Table 1 ijerph-17-05189-t001:** General and sleep characteristics in 18- and 42-month-old children.

	18M Group *	42M Group ^†^
	*n* = 183	*n* = 215
Sex (boy/girl)	93/90	104/111
Firstborns (*n*/%)	84 (45.9%)	91 (42.3%)
Respondent (mother/father)	179/4	207/8
Age (months)	18 (17–19)	42 (42–43)
Sleep duration		
Weekdays (minutes)	720 (660–720)	630 (600–660)
Weekends (minutes)	720 (660–720)	630 (600–660)
Weekend–weekday difference (minutes)	0 (0–0)	0 (0–0)
Bedtime		
Weekdays	9:00 PM (9:00–9:00)	9:00 PM (8:30–9:30)
Weekends	9:00 PM (8:30–9:30)	9:30 PM (9:00–10:00)
Weekend–weekday difference (minutes)	0 (0–30)	0 (0–30)
Wake time		
Weekdays	7:00 AM (6:30–7:27)	7:00 AM (6:30–7:30)
Weekends	7:00 AM (7:00–8:00)	7:30 AM (7:00–8:00)
Weekend–weekday difference (minutes)	0 (0–30)	30 (0–60)
Naps		
Days with naps /week	7 (7–7)	5 (2–6)
Nap duration (minutes)	90 (60–120)	90 (60–120)

Median (25, 75 percentiles); 18M *, 18-month-old group; 42M ^†^, 42-month-old group.

**Table 2 ijerph-17-05189-t002:** Electronic media usage in 18- and 42-month-old children.

Electronic Media Usage (*n*/%)	18M Group *	42M Group ^†^
Television	139 (76.8%)	175 (84.5%)
Laptop/computer	9 (5.0%)	14 (6.8%)
Tablet device	18 (10.2%)	38 (18.6%)
Music player	1 (0.6%)	1 (0.5%)
Smartphone	39 (21.8%)	94 (45.6%)
Mobile phone	1 (0.6%)	0 (0.0%)
Portable game	5 (2.9%)	18 (8.8%)
Home video game	2 (1.1%)	22 (10.8%)

18M ***,** 18-month-old group; 42M **^†^**, 42-month-old group.

**Table 3 ijerph-17-05189-t003:** Comparison of sleep habits in 18-month-old children.

	Nursery School Group (*n* = 79)	Home Group (*n* = 104)	*p*-Value *
Age (months)	18.0 (18.0–19.0)	18.0 (17.0–19.0)	0.185
Sex (boys/girls)	34/45	59/45	0.066
Sleep duration			
Weekdays (minutes)	690 (660–720)	720 (660–750)	0.060
Weekends (minutes)	660 (660–720)	720 (660–750)	0.023 ^†^
Weekend–weekday difference (minutes)	0	0	0.572
Bedtime			
Weekdays	9:00 PM (8:30–9:00)	9:00 PM (8:00–9:15)	0.796
Weekends	9:00 PM (8:30–9:30)	9:00 PM (8:30–9:30)	0.304
Weekend–weekday difference (minutes)	0 (0–30)	0	0.084
Wake time			
Weekdays	6:30 AM (6:10–7:00)	7:00 AM (6:40–7:30)	<0.001 ^‡^
Weekends	7:00 AM (6:30–8:00)	7:00 AM (7:00–8:00)	0.303
Weekend–weekday difference (minutes)	30.0 (0–60)	0 (0–30)	<0.001 ^‡^
Naps			
Days with naps/week	7.0 (7.0–7.0)	7.0 (7.0–7.0)	0.080
Nap duration (minutes)	120 (90–120)	90 (60–120)	0.002 ^†^

Median (25, 75 percentiles); * Chi-square test for age and Mann-Whitney U-test for sleep habits; **^†^**
*p* < 0.05; **^‡^**
*p* < 0.01.

**Table 4 ijerph-17-05189-t004:** Comparison of electronic media usage in 18-month-old children.

Electronic Media Usage (*n*/%)	Nursery School Group (*n* = 79)	Home Group (*n* = 104)	*p*-Value *
Television	61 (78.2%)	78 (75.7%)	0.696
Laptop/computer	6 (7.7%)	3 (2.9%)	0.177
Tablet device	9 (11.7%)	9 (9.1%)	0.622
Music player	0 (0.0%)	1 (1.0%)	1.000
Smartphone	21 (26.9%)	18 (17.8%)	0.150
Mobile phone	0 (0.0%)	1 (1.0%)	1.000
Portable game	0 (0.0%)	5 (5.2%)	0.068
Home video game	1 (1.3%)	1 (1.0%)	1.000

* Exact test.

**Table 5 ijerph-17-05189-t005:** Comparison of sleep habits in 42-month-old children.

	Kindergarten Group	Nursery School Group	Home Group	*p*-Value *	*p*-Value ^†^
	(A)	(B)	(C)		
	*n* = 66	*n* = 114	*n* = 35		
Age (months)	42 (42–42)	42 (41–43)	42 (41–42)	0.031	
Sex (boys/girls)	33/33	62/52	16/19	0.635	
Sleep duration					
Weekdays (minutes)	630 (600–660)	660 (600–690)	600 (600–660)	0.024	
Weekends (minutes)	645 (600–660)	660 (600–660)	600 (600–660)	0.281	
Weekend—weekday difference (minutes)	0 (0–30)	0 (30–0)	0 (0–0)	0.119	
Bedtime					
Weekdays	8:30 p.m.	9:00 p.m.	9:00 PM	<0.001	A&B: < 0.001, A&C: 0.007
	(8:30–9:00)	(9:00–9:30)	(8:30–9:30)		
Weekends	8:30 p.m.	9:30 p.m.	9:00 p.m.	<0.001	A&B: < 0.001
	(8:30–9:00)	(9:00–10:00)	(9:00–10:00)		
Weekend—weekday difference (minutes)					
	0 (0–30)	0 (0–30)	0 (0–30)	0.209	
Wake time					
Weekday	7:00 a.m.	7:00 a.m.	7:30 a.m.	0.011	A&C: 0.021, B&C: 0.013
	(6:30–7:20)	(6:30–7:20)	(6:55–7:30)		
Weekends	7:30 a.m.	7:30 a.m.	8:00 a.m.	0.013	A&C: 0.010
	(7:00–7:45)	(7:00–8:00)	(7:07–8:30)		
Weekend—weekday difference (minutes)					
	30 (0–41.25)	30 (0–60)	30 (0–60)	0.149	
Naps					
Days with naps/week (day)	2 (1–3)	6 (5–7)	3.5 (1.625–5.75)	<0.001	A&B: < 0.001, B&C: < 0.001
Nap duration (minutes)	75 (60–90)	90 (75–120)	60 (52.5–97.5)	0.001	A&B: 0.005, B&C: 0.006

Median (25, 75 percentiles); * Chi–square test for age, * Kruskal–Wallis test and **^†^** Dunn’s test for sleep habits.

**Table 6 ijerph-17-05189-t006:** Comparison of electronic media usage in 42–month–old children.

Electronic Media Usage (*n*/%)	Kindergarten Group	Nursery School Group	Home Group	*p-*Value *
	(A)	(B)	(C)	
	*n* = 66	*n* = 114	*n* = 35	
Television	57 (87.7%)	90 (82.6%)	28 (84.8%)	0.663
Laptop/computer	5 (7.7%)	8 (7.5%)	1 (3.1%)	0.308
Tablet device	16 (25.4%)	18 (16.7%)	4 (12.1%)	0.212
Music player	0 (0.0%)	0 (0.0%)	1 (3.0%)	0.162
Smartphone	31 (47.7%)	50 (46.3%)	13 (39.4%)	0.723
Mobile phone	0 (0.0%)	0 (0.0%)	0 (0.0%)	–
Portable game	12 (18.8%)	5 (4.6%)	1 (3.1%)	0.003 ^†^
Home video game	12 (18.8%)	9 (8.3%)	1 (3.1%)	0.033 ^†^

***** Exact test; ^†^
*p* < 0.05.

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
