# Peer review of "Sleep Habits and Electronic Media Usage in Japanese Children: A Prospective Comparative Analysis of Preschoolers"

_ijerph, 2020, doi:10.3390/ijerph17145189_

Round 1

Reviewer 1 Report

This is an interesting survey of children's sleeping habits, use of electronic media devices which indicates some interesting implications for parents and staff working in kindergartens and nursery schools. I would suggest the following minor changes;

  1. There needs to be a little more detail on the difference between kindergarten and nursery schools as this seems to be key to the study. This would include the type of activities the children engage in there (adult-led or child-led or the balance between the two, physical activity, naps), the length and routine of the day, including obligatory nap times, rather than the detail of the Ministries. 
  2. It would be helpful to know if there is anything done in kindergartens and nursery schools on electronic media usage, including whether children are taught about e-safety or have access to these devices routinely in these early childhood settings.

Author Response

We thank the Reviewer for the valuable comments. We have revised the manuscript according to the Reviewer’s comments.
Please see the attachment.

Reviewer 2 Report

This is a well-written manuscript that explores the difference in sleep and electronic media usage between Japanese children who attend childcare or stay home at ages 18 and 42 months. The research question is an interesting twist on the usual analysis of how electronic media use affects sleep, instead examining whether childcare attendance may be a common cause. Their findings do not strongly support this idea.

Introduction

It would be helpful to the reader to move the description of Japanese nursery school vs. kindergarten to the Introduction from the Discussion, as this is necessary for contextualizing the authors' findings.  Do children who begin in nursery school generally remain in nursery school until they are 5, or do some children switch from nursery school to kindergarten when they are age >3 years?  

Methods

The statistical analysis section is problematic:

Did the authors check the normality of the distributions of their outcome variables?  This has a direct impact on the appropriateness of their choice of statistical methods.  For example, a nonparametric test (Mann-Whitney U test) was used to examine relations with both sleep and media use among 18-month-olds, but a parametric test (ANOVA) was used to examine relations with sleep among 42-month-olds and a non-parametric test (Kruskal-Wallis) was used to examine relations with media use among 42-month-olds.  Also, the authors say they used an ANOVA to compare sex and schooling among 42-month-olds, but these are both categorical variables, so an ANOVA would not work--did they use a chi-square test?  Finally (and very importantly), for any of the continuous variables that were not normally distributed, the tables should present their measures of central tendency as medians, not means.

Results

As mentioned above, it is not clear that means are the appropriate descriptive statistics to be presented for all variables in the tables.  Not knowing whether the appropriate statistical tests were used also makes it difficult to know whether to trust the conclusions drawn from these results.

In Table 5, the authors have used a Bonferroni-corrected p-value to determine statistical significance.  It is unclear why they feel the need to do so only in this table (sleep outcomes in 42-month-olds) when they do not do so for either outcome in 18-month-olds or for media use outcomes in 42-month-olds.  Indeed, if the sleep variables are not entirely independent, an argument could be made that Bonferroni correction is unnecessary even in this case, as there are not multiple "effective" comparisons.  Even so, the presentation of the p-values is unconventional and confusing.  The authors should present the adjusted p-value (0.05/# comparisons) in a footnote and list the conventional p-values in the table.  They can indicate with a symbol which of the p-values meet the stricter cutoff.  Also, which test was used to determine the p-values in the right-most column of the table?  Tukey?  Should be noted in the Methods and in the table itself.  The authors could reduce the number of footnotes by simply listing three categories: statistically significant difference between A & B, B & C, and A & C.  The reader can easily determine which is higher/lower.  Finally, why is there not a similar column in Table 6?

Although statistically significant, it is not clear that a 1-minute difference in portable game use among 18-month-olds is meaningful, especially since only 5 children from the at-home group reported any use at all.  In cases where absolute differences are so small, the authors might want to consider not only the statistical significance of their results, but the real-world meaningfulness of them, as well.  Where there are sparse data, the authors should also be alert to the potential for bias.

In regard to the point the authors make in the Introduction, that children's sleep habits are formed at an early age, it might be interesting to compare sleep outcomes among those who attended any school at age <3 years, those who attended school beginning at >3 years, and those who never attended nursery/kindergarten, if those data are available.

Discussion

As mentioned, the information about Japanese childcare/schooling should be moved to the Introduction.  Here it would be appropriate to delve a little deeper into the potential sociodemographic differences between groups.  For example, how costly is it to send a child to nursery school or kindergarten?  And is there a difference in cost between the two?  Is there social stigma against putting a child in nursery school?  Is keeping a child home considered a privilege?  Examining these factors in more detail would go a long way toward mitigating the very substantial limitation of not having available SES data in the sample.

The point about the shorter kindergarten day leaving more time for media use is well taken.

The discussion about excessive screen time (beginning line 254) would be strengthened if the authors had analyzed whether schooling predicted excessive screen time, perhaps defined according to the AAP recommendation.

Overall, this is an interesting topic and a thoughtful presentation that makes some interesting connections between early childhood life patterns and sleep/media use, but the problems with their statistical methods severely undermine their cause.

Author Response

(The authors gave the same response as above.)

Round 2

Reviewer 2 Report

The authors have thoughtfully revised this manuscript.  There are just a  few minor points that still need to be addressed.  A number of them have to do with discrepancies between the revised data in the tables and the text, which should be updated to match.

Lines 20 and 294, and Table 3: The information on weekend sleep duration in Table 3 doesn't make sense: the nursery and home groups have the same weekend bedtimes and wake times; how can they have different sleep durations?  Please double check the data and, if necessary, adjust the conclusions on lines 20 and 294.

Line 26: remove the second (added) comma.

Lines 59-60: This sentence can be removed, as it appears to say the same thing as the prior sentence.

Line 67: Change "caregivers" to "parents/caregivers" so consistent with line 94.

Line 81: Add a paragraph break before "Despite" and combine with the following paragraph.

Line 127 and Tables 5 and 6: Fisher's Exact Test is used for two-level variables (e.g., 2x2 tables).  Because there are 3 levels in the 42-month analysis (kindergarten, nursery school, and home), a Chi-square test would be more appropriate.

Lines 157, 193, and 252: Change "Main nap duration" to "Nap duration."

Line 220: Tables 5 and 6, not just Table 5.

Table 5: Please check the p-value for Age.  Please check the weekend wake time for home (7:07 seems odd--typo?).  Bonferroni is not a post hoc test.  The appropriate post hoc test would be Dunn's test.  The post hoc test should only be performed when the original p-value is <0.05.  Please be consistent in how you list the comparison groups (A&B, A&C, B&C).

Line 256: Your data show differences in weekday sleep duration among the three groups.

Lines 265 and 267: Please change "means" to "medians."

Lines 266 and 267: Please change "percentiles" to "minutes."

Line 266: Please check "172" to make sure it's not a typo.

Line 296: At 42 months, there was no statistically significant difference in weekday or weekend bedtime between nursery and home children; there was no statistically significant difference in weekday or weekend wake time between nursery and kindergarten children.

Lines 298-9: There was a statistically significant difference in nap duration between nursery and home children at 18 months.

Line 302: Please change "5.6" to "median 6" so matches data presented in Table 5.

Lines 310-11: Compared with nursery school children without naps?  Or do you mean nursery school children, who napped more frequently, had later bedtimes compared with kindergarteners, who napped less frequently?

Line 315: Please change "84.5%" to "85%," as all other percentages are rounded to the nearest whole number.  Insert "most frequently" after "The devices used..."

Lines 331-332: This sentence is not clear.

Lines 332-335: This added section is also hard to understand.  It also belongs in the paragraph above that describes sleep patterns, not in the section on media use.
